# Implementation status of postoperative rehabilitation for older patients with hip fracture in Kyoto City, Japan: A population-based study using medical and long-term care insurance claims data

Kosuke Sasaki[1]*, Yoshimitsu Takahashi[1], Mayumi Toyama[1], Hiroaki Ueshima[2], Tomoko Ohura[1,3], Satoe Okabayashi[4], Tomonari Shimamoto[5], Yukiko Tateyama[5], Hiroko Ikeuchi[5], Junichi Murakami[6], Noriko Furuita[7], Genta Kato[8], Taku Iwami[5], Takeo Nakayama[1]

1 Department of Health Informatics, School of Public Health, Graduate School of Medicine, Kyoto University, Kyoto, Japan, 2 Center for Innovative Research and Education in Data Science, Institute for Liberal Arts and Sciences, Kyoto University, Kyoto, Japan, 3 Evidence-based Long-term Care Team, Center for Gerontology and Social Science, Research Institute, National Center for Geriatrics and Gerontology, Obu, Aichi, Japan, 4 Agency for Health, Safety and Environment, Kyoto University, Kyoto, Japan, 5 Department of Preventive Services, School of Public Health, Graduate School of Medicine, Kyoto University, Kyoto, Japan, 6 Department of Orthopaedic Surgery, Kyoto Min-iren Chuo Hospital, Kyoto, Japan, 7 Department of Obstetrics and Gynecology, Kyoto Min-iren Chuo Hospital, Kyoto, Japan, 8 Department of Hospital Ward Management, Kyoto University Hospital, Kyoto, Japan

* sasaki.kosuke.u73@kyoto-u.jp

**Data Availability Statement:** The data for this study can be obtained from Kyoto City, but access

## Abstract

Continuing rehabilitation after hip fractures is recommended to improve physical function and quality of life. However, the long-term implementation status of postoperative rehabilitation is unclear. This study aims to investigate the implementation status of postoperative rehabilitation for older patients with hip fractures and the factors associated with continuing rehabilitation. A retrospective cohort study evaluated medical and long-term care insurance claims data of patients aged 75 years or older in Kyoto City, Japan, who underwent hip fracture surgeries from April 2013 to October 2018. We used logistic regression analysis to examine factors associated with six-month rehabilitation continuation. Of the 8,108 participants, 8,037 (99%) underwent rehabilitation the first month after surgery, but only 1,755 (22%) continued for six months. The following variables were positively associated with continuing rehabilitation for six months: males (adjusted odds ratio: 1.41 [95% confidence interval: 1.23–1.62]), an intermediate frailty risk (1.50 [1.24–1.82]), high frailty risk (2.09 [1.69–2.58]) estimated using the hospital frailty risk scores, and preoperative care dependency levels: support level 1 (1.69 [1.28–2.23]), support level 2 (2.34 [1.88–2.90]), care-need level 1 (2.04 [1.68–2.49]), care-need level 2 (2.42 [2.04–2.89]), care-need level 3 (1.45 [1.19–1.76]), care-need level 4 (1.40 [1.12–1.75]), and care-need level 5 (1.31 [0.93–1.85]). In contrast, dementia was cited as a disincentive (0.53 [0.45–0.59]). Less than 30% of older patients continued rehabilitation for six months after surgery. Factors associated with

is restricted due to contractual agreements specific to this study, and they are not publicly accessible. This study used secondary, anonymized data and was approved by the Information Disclosure and Personal Information Protection Review Board of Kyoto City and the Ethics Committee of Kyoto University Graduate School and Faculty of Medicine (approval number R2690). Only researchers registered with Kyoto City can use the database.

**Funding:** This study was supported by the operating expenses of the Department of Health Informatics, Graduate School of Public Health, Kyoto University [grant number 021515 to TN]; a grant-in-aid for scientific research (KAKENHI) from the Japan Society for the Promotion of Science [grant numbers 20H01594 to YT and 21K21166 to MT]; the programs for Progress of the next Cross-ministerial Strategic Innovation Promotion Program (SIP) on "Integrated Health Care System (C-1)" from the Cabinet Office, Government of Japan [grant number JPJ012425 to TN]; and a consignment fee from Kyoto City (FY 2020 to TN). This study was conducted as part of a project commissioned by Kyoto City, Japan. The funder did not play any role in the study design, data collection and analysis, decision to publish, or preparation of the manuscript.

**Competing interests:** We have read the journal's policy, and the authors of this manuscript have the following competing interests to declare: This study was conducted as part of a project commissioned by Kyoto City, Japan. TN reported receiving grants from the operating expenses of the Department of Health Informatics, Graduate School of Public Health, Kyoto University [grant number 021515]; from the Cabinet Office, Government of Japan [grant number JPJ012425]; and a consignment fee from Kyoto City (FY 2020). YT and MT reported receiving grants from the Japan Society for the Promotion of Science during the study [grant numbers 20H01594 and 21K21166]. JM received honoraria for lectures from ASAHI KASEI PHARMA CORPORATION and UCB Japan Co. Ltd. outside this submitted work.

continuing rehabilitation were male sex, higher frailty risk, care dependency before hip fracture surgeries, and non-dementia.

## Introduction

Hip fractures represent a profound health challenge, resulting in increased mortality rates [1], diminished activities of daily living (ADLs), and compromised quality of life [2, 3]. As global demographics shift towards an older populace [4], hip fractures are anticipated to emerge as a progressively critical issue, particularly in Japan, which is recognized for its unparalleled global aging rate [5]. Asia accounts for half of the global incidents of hip fractures [6]. The projected cases are set to rise substantially, from 1.12 million in 2018 to 2.56 million by 2050 [7]. Specifically in Japan, there has been a notable escalation in hip fractures, with an approximately 2.5-fold increase from 76,600 cases in 1992 to 193,400 cases in 2017 [8]. Fractures constitute 12% of the causes of the need for long-term care (LTC), ranking as the third most prevalent cause in Japan [9].

Clinical practice guidelines from nations including Australia [10], the United Kingdom [11], the United States [12, 13], Korea [14], and Japan [15, 16] endorse early postoperative rehabilitation for managing hip fractures and continuing post-hospitalization. Although the appropriate duration of rehabilitation varies depending on the healthcare system and the patient's condition, there are reports that rehabilitation for 3 to 6 months is necessary after hip fracture surgeries. A randomized controlled trial of older patients with hip fractures found that outpatient rehabilitation using progressive resistance training for six months after discharge significantly improved walking ability and quality of life compared to low-intensity home exercises [17]. A prospective cohort study found that most aspects of the SF-36, except for the physical role behavior subscale, showed almost complete recovery six months after hip fracture surgeries [18]. Such consistent rehabilitation post-hip surgery enhances physical function [19–21] and quality of life [19, 20]. The Japanese guidelines specifically recommend a minimum of six months of postoperative rehabilitation [15, 16]. In Japan, the average length of hospital stay is decreasing, making rehabilitation after discharge from acute care hospitals increasingly important [22]. In Japan's multifaceted treatment approach, patient discharge trajectories differ, spanning transfers to other facilities and direct-to-home discharge [23–25]. Uninterrupted rehabilitation is crucial, regardless of transitions in the care setting.

It has been suggested that there are disparities in access to rehabilitation based on patient characteristics. A systematic review describing the equity in randomized controlled trials of rehabilitation interventions following hip fractures revealed that, in > 50% of the 35 trials, potential participants were systematically excluded based on conditions such as residing in nursing homes, cognitive impairment, mobility or functional impairments, age, and ineligibility surgery [26]. A retrospective cohort study using the census database of all inpatient services provided by public and private hospitals in New South Wales, Australia, reported that older patients, women, those with dementia, those with high frailty risk, and those residing in LTC facilities were less likely to receive in-hospital rehabilitation [27, 28]. Another retrospective cohort study that use of the Diagnosis Procedure Combination database, which collects inpatient data from over 1,500 hospitals across Japan, reported that almost all patients who underwent hip fracture surgeries in acute care hospitals received short-term rehabilitation [29, 30]. However, it also reported that those with dementia had fewer opportunities for inpatient rehabilitation after discharge from acute care hospitals [29]. The rate of mid- to long-term

extension of rehabilitation after discharge from acute care hospitals for patients following treatments for hip fractures has not been previously reported on, which has raised some concerns surrounding whether sufficient rehabilitation is being provided. Hence, this study aimed to explore the current status of rehabilitation implementation, and identify the factors influencing sustained rehabilitation in older patients with hip fractures post-discharge in Japan—a nation that is currently facing the challenges associated with a rapidly-aging population.

## Materials and methods

### Study design and data source

This retrospective cohort study used databases such as medical insurance claims data in Kyoto City [31] Kyoto City collects the data, including the specific health check-ups and specific health guidance data, the medical insurance claims data, and LTC insurance claims data to understand the situation of disease incidence among citizens and the status of prevention, treatment, and LTC, as well as to build evidence that can be utilized in policies. In this study, we used data spanning April 2013 to March 2019 from the following sources: the Japanese "Medical Care System for the Elderly in the Latter Stage of Life" (which is the mandatory medical insurance system for individuals > 75 years of age) [32] basic resident registration, medical insurance claims data, LTC insurance claims data, and LTC needs certification data. These data were merged individually using a unique identifier by Kyoto City and anonymized before we received the data. The proportions of personal identification numbers assigned in this database were as follows. Personal identification numbers were assigned to ~99% of the medical claims data for injury and illness codes (385,109,527 of 387,547,567 records), ~99% of the medical activity codes (1,237,350,713 of 1,253,415,997 records), ~98% of the LTC insurance claims data (52,387,340 of 53,494,825 records), and ~89% of the LTC care-requiring certification data (861,376 of 964,429 records). This study was conducted by the secondary use of existing anonymized data and approved by the Information Disclosure and Personal Information Protection Review Board of Kyoto City and the Ethics Committee of Kyoto University Graduate School and Faculty of Medicine (approval number was R2690). We first accessed the database for research purposes on December 10, 2020. Only Kyoto City could access personal information. Signed informed consent was waived because the researchers received anonymized existing data from Kyoto City.

### Participants

The participants of this study were patients aged 75 years or older who underwent surgery for hip fractures from April 2013 to October 2018 among the participants in the Medical Care System for the Elderly in the Latter Stage of Life in Kyoto. We included patients aged 75 years or older because we used the data of the Medical Care System for the Elderly in the Latter Stage of Life, which is the medical insurance system for all people over 75 years old [32]. The extracted codes were "fracture of neck of femur" (The International Statistical Classification of Diseases and Related Health Problems Tenth Revision [ICD-10]: S72.0) and "pertrochanteric fracture" (ICD-10: S72.1), excluding cases of "suspected fracture." The surgical procedures for hip fractures were defined according to the medical treatment codes (S1 Table). The following patients were excluded: (1) patients under 75 years old; (2) patients with "greater trochanteric fracture" (ICD-10: S72.1) [33]; (3) patients who were "transferred," "died," "transferred out of the country," or "disappeared" within six months of the surgery for hip fracture, according to the information in the Basic Resident Registration. Only the first surgery was considered in this study when the patients underwent multiple surgeries during the study period. The participants were identified on the advice of two orthopedists (HI and JM).

## Primary outcome

The primary outcome was the implementation of rehabilitation for six months after hip fracture surgery as a process measure. We defined the use of rehabilitation as rehabilitation-related claims filed under medical or LTC insurance at least once per month following hip fracture surgery (S2 and S3 Tables). In Japan, there are two types of insurance-covered rehabilitation: medical and LTC. Generally, acute and restorative rehabilitation is covered by medical insurance, while chronic rehabilitation is covered by both insurance types. This distinction ensures that patients receive appropriate care tailored to their stage of recovery and long-term needs [24]. To use LTC insurance services, patients must be certified based on their physical and cognitive functions, as well as their care and medical needs. Once certified, patients are required to use LTC insurance rehabilitation services, except during hospitalization or the initial recovery phase [34]. Because this study aimed to investigate the implementation of mid- to long-term rehabilitation after hip fractures, we included both medical and LTC insurance data. The definition of implementing rehabilitation was defined based on the advice of a physiatrist (MT), physical therapists (KS and HU), and an occupational therapist (TO).

## Measurements

The baseline variables at the hip fracture surgery included age, sex, type of fracture, type of surgery for hip fracture, comorbidities (dementia, depression, and delirium), frailty, 20 types of diseases (S4 Table) [35] the level of LTC need before the surgery, and the place where the survey on certification of LTC need was conducted.

As comorbidity, dementia [36] depression [37, 38] and delirium [39, 40] were identified using the ICD-10 codes (S4 Table). Frailty was estimated using the hospital frailty risk score [40] The hospital frailty risk score was calculated by assigning points to 109 ICD-10 codes and was classified into low ($<$5), intermediate (5–15), and high ($>$15) [40] Dementia, depression, and the hospital frailty risk score were identified if the medical treatment was given within the last 12 months from the month of surgery. Delirium was recognized if the medical treatment was given within six months from the month of surgery. The 20 types of diseases [35] were identified if the medical treatment was provided within six months after the surgery. If one or more of the 20 specified disease types are applicable, a patient's rehabilitation status may not be confirmed by this database, because claims related to some types of rehabilitation are made on paper. Therefore, we specifically identified these 20 disease types to account for the possibility of underestimating the implementation of rehabilitation. Comorbidities, including the hospital frailty risk scores and the 20 different diseases, were excluded for "suspected diseases." For the level of the LTC required and the place where the certification of LTC need was conducted, certification before surgery for hip fractures and within the valid period of certification in the month of surgery was considered. Depending on the subject's physical and mental status and the estimated hours of care per day, LTC care needs are classified into seven levels: support levels 1 and 2 and care-need levels 1 to 5, with higher levels indicating higher care needs. For example, people with care-need level 5 require constant care [41] The latest one was used if the LTC needs was certified multiple times before hip fracture surgeries.

We also extracted the status of admission to the convalescent rehabilitation ward and the usage of the discharge support service from the month of surgery onward. The convalescent rehabilitation ward is a ward for the intensive rehabilitation of patients who require ADLs assistance—including those with hip fracture or cerebrovascular disease—to prevent bedridden patients from staying bedridden, and help them return to their homes by improving their ADLs [42]. We identified the admission to the convalescent rehabilitation ward within three months after surgery (medical treatment code: A308-00). We identified the usage of discharge

support services to monitor the discharge status. We defined discharge support service as calculating the items within six months after surgery (S5 Table).

## Statistical analysis

Continuous variables were shown as a median and interquartile range. Categorical variables were shown as frequencies and percentages. The proportion of patients receiving postoperative rehabilitation was calculated each month up to six months after surgery. Additionally, the proportion of rehabilitation was calculated with stratification by medical insurance (inpatient or outpatient) or LTC insurance. The percentage of patients who continued rehabilitation under medical insurance (inpatient or outpatient) or LTC insurance for six months after surgery was also calculated. Multivariable logistic regression was used to explore the factors associated with continuing rehabilitation for six months after surgery as the dependent variable. The explanatory variables in the model included age groups (75–84 years old, 85–94 years old, or 95 years or older), sex, type of fracture (fracture of head and neck of femur, or pertochanteric fracture), dementia (yes or no), depression (yes or no), delirium (yes or no), the hospital frailty risk score (low, intermediate, or high risk), and care-level at the surgery for hip fractures (no certification, support level 1, support level 2, care-need level 1, care-need level 2, care-need level 3, care-need level 4, or care-need level 5). Adjusted odds ratios and 95% confidence intervals were calculated. Stata® 17.0 (StataCorp LLC, College Station, Texas, the United States) was used for the analysis.

## Results

The study included 8,108 participants (Fig 1). The median age of the participants was 86 years, with an interquartile range (IQR) of 82–91 years. Of the 8,108 subjects analyzed, 6,736 (83.1%) were women. Of the 8,108 analyzed participants, 5,963 (73.5%) were certified for LTC before surgery (Table 1).

Rehabilitation implementation under either medical insurance (inpatient and outpatient) or LTC insurance decreased from 99% to 28% from months one to six post-surgery (Fig 2). S1 Fig showed the implementation status of rehabilitation for six months after surgery, stratified by medical and LTC insurance.

Overall, 21.6% continued rehabilitation for six months, with <1% never participating under medical or LTC insurance.

By three months, 42.4% had been admitted to a convalescent rehabilitation ward, and 55.3% had received discharge support within six months.

Factors positively influencing six-month rehabilitation continuation were male gender, higher hospital frailty risk scores, and care dependency before surgery (Table 2). Dementia was negatively associated with continuing rehabilitation (Table 2). Women, across all age categories, were less likely than men to continue rehabilitation for six months (S6 Table).

## Discussion

In this population-based study, most patients aged 75 and over in Kyoto City, Japan, underwent rehabilitation in the month following surgery, aligning with prior research [29] However, participation dwindled to below 30% by the sixth month. Despite clinical guidelines advocating for ongoing postoperative rehabilitation, a gap persisted between evidence and practice in sustaining post-surgical rehabilitation. This research's strength lies in illuminating the long-term rehabilitation trajectory utilizing integrated data, encompassing medical and LTC insurance claims.

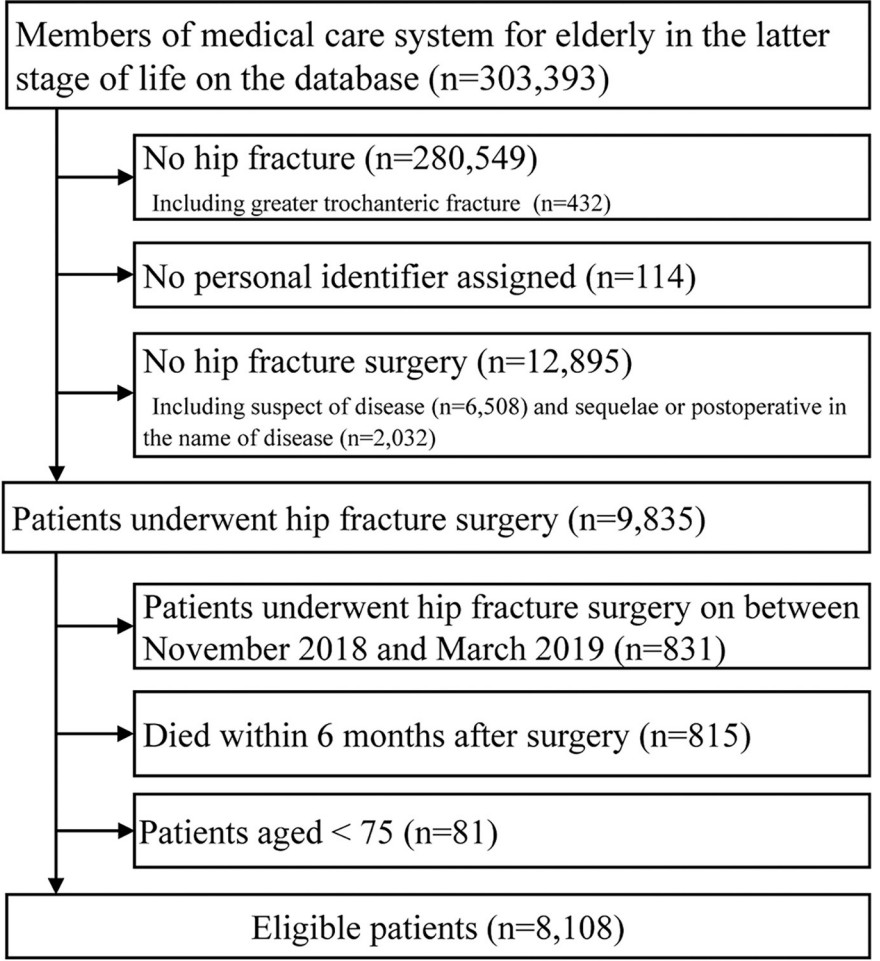

**Fig 1. Flow diagram of the patient selection process.**

This study showed that most patients received postoperative rehabilitation during the inaugural month. However, the number of patients receiving rehabilitation decreased to less than 30% by the sixth month. Transition points in care might account for this cessation of rehabilitation. A systematic review showed the inadequate communication and data sharing between hospital and primary-care physicians upon patient discharge, which could disrupt subsequent care [43] Collaborative engagement with medical institutions and LTC services is essential at discharge. A previous Japanese study using medical and LTC insurance data demonstrated the effectiveness of rehabilitation within one month of discharge to prevent a decline in the level of care for older patients [44]. Establishing a system for a seamless transition to post-acute rehabilitation care is critical.

Other factors should also be considered when attempting to explain the decline in the rehabilitation implementation rate over time after surgery. First, some patients may have experienced relatively rapid recovery in terms of ADLs and physical function following hip fracture surgery, and thus no longer required rehabilitation. Second, some patients may have been reluctant to continue rehabilitation for various reasons. A previous study reported that feelings of vulnerability in patients with hip fractures persist until their function and activities are sufficiently restored to allow them to participate in meaningful activities [45]. Patients who feel

**Table 1. Baseline characteristics of older patients with hip fracture (n = 8,108).**

| Variables | All (n = 8,108) | | Continuing rehabilitation for six months after surgery | | | |
|---|---|---|---|---|---|---|
| | | | Yes (n = 1,755) | | No (n = 6,353) | |
| Age in years, median (IQR) | 86 | [82–91] | 86 | [83–90] | 86 | [82–91] |
| Age groups, n (%) | | | | | | |
| 75–84 years old | 3,091 | (38.1) | 660 | (37.6) | 2,431 | (38.3) |
| 85–94 years old | 4,254 | (52.5) | 943 | (53.7) | 3,311 | (52.1) |
| 95 years or older | 763 | (9.4) | 152 | (8.7) | 611 | (9.6) |
| Sex, women, n (%) | 6,736 | (83.1) | 1,388 | (79.1) | 5,348 | (84.2) |
| Type of fracture, n (%) | | | | | | |
| Fracture of head and neck of femur | 4,508 | (55.6) | 956 | (54.5) | 3,552 | (55.9) |
| Pertrochanteric fracture | 3,600 | (44.4) | 799 | (45.5) | 2,801 | (44.1) |
| Surgical procedure, n (%) | | | | | | |
| Open reduction internal fixation | 4,474 | (55.2) | 957 | (54.5) | 3,517 | (55.4) |
| Open reduction internal fixation for intra-articular fracture | 572 | (7.1) | 105 | (6.0) | 467 | (7.4) |
| Arthroscopic open reduction internal fixation for intra-articular fracture | 0 | (0.0) | 0 | (0.0) | 0 | (0.0) |
| Bipolar hip arthroplasty | 2,972 | (36.7) | 667 | (38.0) | 2,305 | (36.3) |
| Total hip arthroplasty | 90 | (1.1) | 26 | (1.5) | 64 | (1.0) |
| Comorbidity, n (%) | | | | | | |
| Dementia | 3,126 | (38.6) | 546 | (31.1) | 2,580 | (40.6) |
| Depression | 1,213 | (15.0) | 278 | (15.8) | 935 | (14.7) |
| Delirium | 613 | (7.6) | 130 | (7.4) | 483 | (7.6) |
| Hospital frailty risk score, n (%) | | | | | | |
| Low risk (<5) | 1,011 | (12.5) | 156 | (8.9) | 855 | (13.5) |
| Intermediate risk (5–15) | 4,477 | (55.2) | 951 | (54.2) | 3,526 | (55.5) |
| High risk (>15) | 2,620 | (32.3) | 648 | (36.9) | 1,972 | (31.0) |
| Diseases specified by the Minister of Health, Labour and Welfare, n (%) | 842 | (10.4) | 246 | (14.0) | 596 | (9.4) |
| Care-level at surgery for hip fracture, n (%) | | | | | | |
| Not certified | 2,145 | (26.5) | 337 | (19.2) | 1,808 | (28.5) |
| Requiring help 1 | 341 | (4.2) | 86 | (4.9) | 255 | (4.0) |
| Requiring help 2 | 570 | (7.0) | 182 | (10.4) | 388 | (6.1) |
| Long-term care level 1 | 935 | (11.5) | 244 | (13.9) | 691 | (10.9) |
| Long-term care level 2 | 1,517 | (18.7) | 441 | (25.1) | 1,076 | (16.9) |
| Long-term care level 3 | 1,467 | (18.1) | 265 | (15.1) | 1,202 | (18.9) |
| Long-term care level 4 | 859 | (10.6) | 155 | (8.8) | 704 | (11.1) |
| Long-term care level 5 | 274 | (3.4) | 45 | (2.6) | 229 | (3.6) |
| Place of Certification of Needed Long-Term Care survey before surgery, n (%) | | | | | | |
| Home | 3,884 | (47.9) | 1,103 | (62.8) | 2,781 | (43.8) |
| Nursing home | 1,337 | (16.5) | 100 | (5.7) | 1,237 | (19.5) |
| Medical institutions | 743 | (9.2) | 216 | (12.3) | 527 | (8.3) |
| No application | 2,144 | (26.4) | 336 | (19.1) | 1,808 | (28.4) |

IQR: interquartile range

burdened or lack confidence in their recovery are more likely to refuse rehabilitation. Third, some patients may have prioritized services other than rehabilitation. A population-based retrospective cohort study showed that older adults with dementia used home care services more often after sustaining hip fractures, compared to their counterparts without dementia [46]. Additionally, it is necessary to consider the possibility that rehabilitation resources are limited

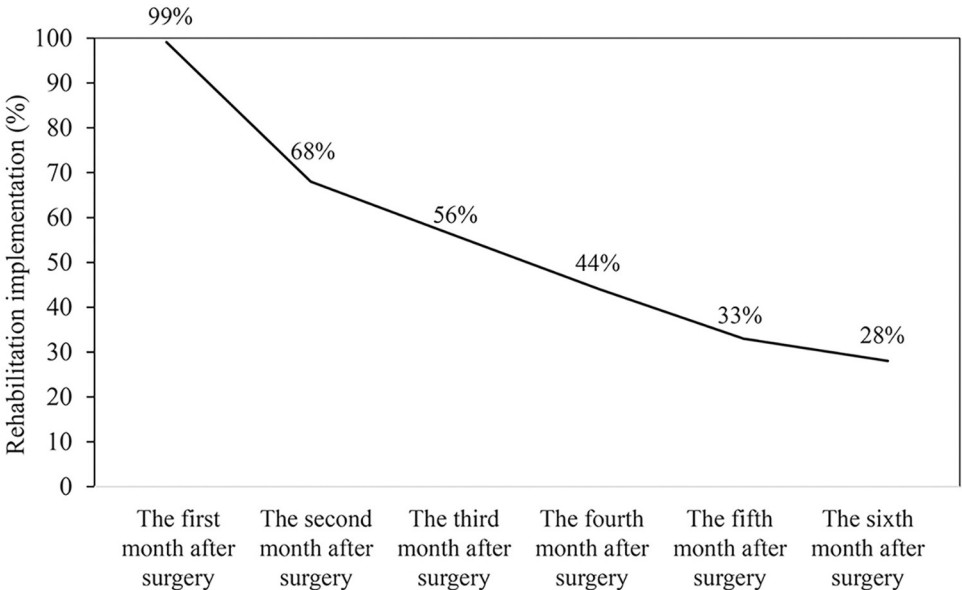

**Fig 2. Implementation status of rehabilitation for six months after surgery (n = 8,108).**

to the patients' residential areas. A study using the National Administrative Database of Canada compared the discharge destinations of patients with hip fractures from acute care hospitals by province, and reported significant variations in discharge destinations across them [47]. Our study could not elucidate the reasons for the decline in the rehabilitation implementation rate over time after surgery. However, future research should consider patient conditions, preferences, and available rehabilitation resources.

Over half the participants had not been admitted to the convalescent rehabilitation ward three months post-surgery. Japan's Clinical Pathway with Regional Alliance system aims to facilitate patients' return home. This system allows multiple facilities to coordinate a common medical pathway that guides patients from acute care facilities through convalescent rehabilitation facilities [48, 49]. In Japan, hip fracture is one of the target diseases for admission to convalescent rehabilitation wards, highlighting the need for intensive rehabilitation. A previous study using the nationwide inpatient database for acute care hospitals in Japan suggested that patients with dementia following hip fractures were often discharged to non-medical settings, with fewer opportunities for hospitalization for rehabilitation [29] Our study provides a deeper understanding of patient transfers to these rehabilitation wards, supplementing prior nationwide data. Exploring further factors dictating discharge locations is crucial, including patient-specific and systemic variables.

Consistent with the previous study [28], this study found that dementia was a factor that inhibited continuing postoperative rehabilitation. Barriers to patients with hip fractures and dementia receiving rehabilitation include inadequate hospital information management, patient prioritization, ineffective comorbidity management, staff education deficiencies, community perceptions, and resource constraints [50]. The multidisciplinary consensus on the care for older patients with hip fractures recommended that patients with dementia after hip fracture surgeries receive the same rehabilitation as those without dementia [51]. A previous study suggested that rehabilitation for patients with dementia after hip fractures improved motor function and ADLs [28] Additionally, intensive inpatient rehabilitation might improve the physical function of older patients with cognitive impairment after hip fractures [52].

**Table 2. Multivariable logistic regression analysis for factors associated with continued rehabilitation for six months after surgery (n = 8,108).**

|  | Univariable | | | Multivariable[a] | | |
|---|---|---|---|---|---|---|
|  | Crude OR | 95% CI | *p*-value | Adjusted OR | 95% CI | *p*-value |
| Age groups |  |  |  |  |  |  |
| 75–84 years old | Reference |  |  | Reference |  |  |
| 85–94 years old | 1.05 | 0.94–1.17 | 0.404 | 1.01 | 0.90–1.14 | 0.855 |
| 95 years or older | 0.92 | 0.75–1.12 | 0.385 | 0.88 | 0.71–1.08 | 0.219 |
| Sex |  |  |  |  |  |  |
| Women | Reference |  |  | Reference |  |  |
| Men | 1.41 | 1.23–1.61 | <0.001 | 1.41 | 1.23–1.62 | <0.001 |
| Type of fracture |  |  |  |  |  |  |
| Fracture of head and neck of femur | Reference |  |  | Reference |  |  |
| Pertrochanteric fracture | 1.06 | 0.95–1.18 | 0.283 | 1.07 | 0.96–1.20 | 0.223 |
| Comorbidity (versus no diagnosis) |  |  |  |  |  |  |
| Dementia | 0.66 | 0.59–0.74 | <0.001 | 0.53 | 0.45–0.59 | <0.001 |
| Depression | 1.09 | 0.94–1.26 | 0.243 | 1.01 | 0.86–1.17 | 0.946 |
| Delirium | 0.97 | 0.79–1.19 | 0.784 | 0.95 | 0.77–1.17 | 0.639 |
| Hospital frailty risk score |  |  |  |  |  |  |
| Low | Reference |  |  | Reference |  |  |
| Intermediate | 1.48 | 1.22–1.78 | <0.001 | 1.50 | 1.24–1.82 | <0.001 |
| High | 1.80 | 1.49–2.18 | <0.001 | 2.09 | 1.69–2.58 | <0.001 |
| Care-level before surgery |  |  |  |  |  |  |
| No certification | Reference |  |  | Reference |  |  |
| Support level 1 | 1.79 | 1.36–2.36 | <0.001 | 1.69 | 1.28–2.23 | <0.001 |
| Support level 2 | 2.48 | 2.01–3.07 | <0.001 | 2.34 | 1.88–2.90 | <0.001 |
| Care-need level 1 | 1.89 | 1.56–2.28 | <0.001 | 2.04 | 1.68–2.49 | <0.001 |
| Care-need level 2 | 2.24 | 1.90–2.63 | <0.001 | 2.42 | 2.04–2.89 | <0.001 |
| Care-need level 3 | 1.22 | 1.02–1.45 | 0.028 | 1.45 | 1.19–1.76 | <0.001 |
| Care-need level 4 | 1.22 | 0.99–1.50 | 0.057 | 1.40 | 1.12–1.75 | 0.003 |
| Care-need level 5 | 1.17 | 0.84–1.62 | 0.362 | 1.31 | 0.93–1.85 | 0.122 |

OR: odds ratio, CI: confidence interval

[a]Model included age groups (75–84 years old, 85–94 years old, or 95 years or older), sex, type of fractures (fractures of head and neck of femur, or pertochanteric fractures), dementia diagnosis (yes or no), depression diagnosis (yes or no), delirium diagnosis (yes or no), hospital frailty risk score (low, intermediate, or high) and care-level at the surgeries for hip fractures (no certification, support level 1, support level 2, care-need level 1, care-need level 2, care-need level 3, care-need level 4, or care-need level 5).

Thus, the diagnosis of dementia should inform, rather than preclude, personalized, ongoing rehabilitation strategies.

In this study, Male sex was a positively associated factor for continuing postoperative rehabilitation. Disparities based on sex may be evident in access to health care [53, 54]. A previous study has shown that women are less likely to be referred for outpatient cardiac rehabilitation [55], while men are less likely to receive recommended hip fracture care, including post-discharge rehabilitation [56]. Interviews with older women 12 months after a hip fracture showed that self-efficacy supported their participation in exercise [57]. Another study focusing on community-dwelling older adults, particularly those with hip fractures, found that men had greater exercise self-efficacy [58]. The Japanese Survey on Time Use and Leisure Activities found that women aged 75 and older spent about 2.4 times more hours per week on household chores than their male counterparts [59]. This may indicate that women are more likely than

men to be reintegrated into society, reducing the need for further rehabilitation. The precise mechanism of this pattern remains unclear; however, the factors of exercise self-efficacy and multiple roles might be intertwined with these observed gender disparities.

This study showed that a high risk of frailty was a factor that promotes continued rehabilitation after surgery. A systematic review identified frailty and reduced grip strength as predictors of adverse functional outcomes in patients with hip fractures [60]. Rehabilitation may be prescribed more consistently for high-risk patients who have a greater need for rehabilitation.

In this study, we found preoperative care dependency was associated with continued rehabilitation after surgery. We infer two aspects of this finding. First, patients who were dependent on care preoperatively could have been likely considered in greater need of postoperative rehabilitation. This is supported by previous research showing a strong correlation between care needs and the Barthel Index [61] with preoperative physical function and ADLs in older patients with hip fractures associated with their functional recovery post-discharge [62]. Patients with lower ADLs and physical function might have been considered in need of rehabilitation and continued rehabilitation. This study also showed that those classified as "support level 2" and "care-need levels 1 or 2" were more inclined to continue with rehabilitation. Generally, individuals with "support needs" can perform most basic ADLs independently, and those with "care needs 1 or 2" require some nursing care. By contrast, those with "care-need level 3" and above require almost total nursing care [63, 64]. Although this study did not investigate the provision of care services other than rehabilitation, it is possible that rehabilitation was prioritized over other services for "care-need level 2" and below. Second, patients who required LTC before surgery were more likely to have established support systems and may have had more accessible access to seamless post-discharge services, including rehabilitation. The local municipal office typically issues a care needs certification notice within 30 days following the application [65]. As a result, patients who were certified as needing care before surgery might experience more seamless access to post-discharge services, including rehabilitation.

This study was subject to several key limitations worth noting. First, we did not assess patient status—including physical function and ADLs, as well as knowledge and attitudes in patients and their families—or the post-discharge environment. Some patients may have discontinued rehabilitation after surgery because their physical function and abilities to perform ADLs improved sufficiently. Second, this database does not contain information about the physicians who prescribed rehabilitation and information concerning the hospitals or facilities where the patients stayed is not well organized. Therefore, this study could not clarify the relationship between hospitals, facilities, and physicians with the implementation of rehabilitation. Further research is warranted to explore the perspectives of patients and their families on this matter [66, 67], as well as facility-level [68] and regional-level factors [69]. Third, the appropriate duration of rehabilitation may vary depending on the healthcare system and the patient's condition, and this study was not able to determine whether all patients with hip fractures should undergo rehabilitation for six months. However, it does highlight the evidence-practice gap in the implementation of rehabilitation following hip fracture surgeries in Japan. Fourth, this study did not examine the content, frequency, or amount of rehabilitation. Fifth, if a patient had any of the 20 disease types specified by the Japanese Ministry of Health, Welfare and Labor, some rehabilitation claims were made on paper and may therefore not have been captured in the database that was used for this study. In addition, we excluded the patients who died within six months after surgery. Consequently, we may have underestimated the implementation status of rehabilitation. Finally, this study included only residents of Kyoto City aged 75 or older. Therefore, careful extrapolation is needed for areas with different medical resources, population sizes, and age structures.

## Conclusions

This study suggests that patients with hip fractures may discontinue rehabilitation after discharge from acute care hospitals. Factors such as older age, male sex, frailty, care dependency before hip fracture surgeries, and non-dementia might be positively associated with completing six months of postoperative rehabilitation. Further research is needed to explore unexamined determinants, including ADLs, patient and family perceptions, and regional differences.

## Supporting information

**S1 Fig. Implementation status of rehabilitation for six months after surgery, stratified by medical and long-term care insurance (n = 8,108).** The number of patients in the same month includes those who underwent rehabilitation by medical insurance (inpatient and outpatient) and long-term care insurance. Therefore, per month, the total number of patients in each category may exceed the number in the analysis.
(TIF)

**S1 Table. The list of definitions of the surgical procedures for hip fracture.**
(XLSX)

**S2 Table. The list of definitions of rehabilitation in medical insurance.**
(XLSX)

**S3 Table. The list of definitions of rehabilitation in long-term care insurance.**
(XLSX)

**S4 Table. The list of definitions of comorbidities.** ICD-10: The International Statistical Classification of Diseases and Related Health Problems tenth Revision, SMON: Subacute Myelo-Optico-Neuropathy, †"Mechanically ventilated conditions" were identified by the medical treatment code.
(XLSX)

**S5 Table. The list of definitions of discharge support service.**
(XLSX)

**S6 Table. Status of continuing rehabilitation for six months after surgery, stratified by sex and age groups (n = 8,108).**
(XLSX)

## Acknowledgments

We would like to express our sincere gratitude to Kyoto City, which provided the data used in this research, and all the people at the Graduate School of Public Health, Kyoto University.

## Author Contributions

**Conceptualization:** Kosuke Sasaki, Yoshimitsu Takahashi, Takeo Nakayama.

**Data curation:** Kosuke Sasaki, Yoshimitsu Takahashi.

**Formal analysis:** Kosuke Sasaki, Yoshimitsu Takahashi.

**Funding acquisition:** Yoshimitsu Takahashi, Mayumi Toyama, Takeo Nakayama.

**Investigation:** Kosuke Sasaki.

**Methodology:** Kosuke Sasaki, Yoshimitsu Takahashi, Takeo Nakayama.

**Project administration:** Yoshimitsu Takahashi, Taku Iwami, Takeo Nakayama.

**Resources:** Yoshimitsu Takahashi, Genta Kato, Taku Iwami, Takeo Nakayama.

**Supervision:** Yoshimitsu Takahashi, Taku Iwami, Takeo Nakayama.

**Visualization:** Kosuke Sasaki.

**Writing – original draft:** Kosuke Sasaki.

**Writing – review & editing:** Kosuke Sasaki, Yoshimitsu Takahashi, Mayumi Toyama, Hiroaki Ueshima, Tomoko Ohura, Satoe Okabayashi, Tomonari Shimamoto, Yukiko Tateyama, Hiroko Ikeuchi, Junichi Murakami, Noriko Furuita, Genta Kato, Taku Iwami, Takeo Nakayama.

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
