## [Decision Letter · Decision Letter 0]

21 May 2024

PONE-D-24-11365Implementation status of postoperative rehabilitation for older patients with hip fracture in Kyoto City, Japan: A population-based study using medical and long-term care insurance claims dataPLOS ONE

Dear Dr. Sasaki,

Thank you for submitting your manuscript to PLOS ONE. After careful consideration, we feel that it has merit but does not fully meet PLOS ONE’s publication criteria as it currently stands. Therefore, we invite you to submit a revised version of the manuscript that addresses the points raised during the review process.

We look forward to receiving your revised manuscript.

Kind regards,

Antimo Moretti

Academic Editor

PLOS ONE

Journal Requirements:

   "We have read the journal's policy and the authors of this manuscript have the following competing interests:

This study was conducted as part of a project commissioned by Kyoto City, Japan and supported by a consignment fee from Kyoto City (FY 2020). This work was supported by the operating expenses of the Department of Health Informatics, Graduate School of Public Health, Kyoto University [grant number 021515]. YT and MT reported receiving grants from the Japan Society for the Promotion of Science during the study [grant numbers 20H01594 and 21K21166]. HU reported receiving grants from Janssen Pharmaceutical K.K.; Chugai Pharmaceutical Co Ltd; Mitsubishi Tanabe Pharma Corporation; Sumitomo Pharma Co Ltd; Takeda Pharmaceutical Company Limited; KYOCERA Corporation; Nippon Telegraph and Telephone Data Corporation; Nippon Telegraph and Telephone Business Solutions Corporation; and Health Insurance Claims Review & Reimbursement services outside this submitted work. JM received honoraria for lectures from ASAHI KASEI PHARMA CORPORATION and UCB Japan Co. Ltd. outside this submitted work."

We note that you received funding from a commercial source:  Janssen Pharmaceutical K.K.; Chugai Pharmaceutical Co Ltd; Mitsubishi Tanabe Pharma Corporation; Sumitomo Pharma Co Ltd; Takeda Pharmaceutical Company Limited; KYOCERA Corporation; Nippon Telegraph and Telephone Data Corporation; Nippon Telegraph and Telephone Business Solutions Corporation; and Health Insurance Claims Review,  ASAHI KASEI PHARMA CORPORATION and UCB Japan Co. Ltd. 

3. For studies involving third-party data, we encourage authors to share any data specific to their analyses that they can legally distribute. PLOS recognizes, however, that authors may be using third-party data they do not have the rights to share. When third-party data cannot be publicly shared, authors must provide all information necessary for interested researchers to apply to gain access to the data. (https://journals.plos.org/plosone/s/data-availability#loc-acceptable-data-access-restrictions) 

Reviewers' comments:

Reviewer's Responses to Questions

**Comments to the Author**

1. Is the manuscript technically sound, and do the data support the conclusions?

Reviewer #1: Yes

2. Has the statistical analysis been performed appropriately and rigorously? 

Reviewer #1: No

3. Have the authors made all data underlying the findings in their manuscript fully available?

Reviewer #1: Yes

4. Is the manuscript presented in an intelligible fashion and written in standard English?

Reviewer #1: Yes

5. Review Comments to the Author

Reviewer #1: This study aimed to examine the implementation status of postoperative rehabilitation for older patients with hip fractures and the factors associated with continuing rehabilitation, using Japanese medical and long-term care insurance claims data in Kyoto. The authors found a decreased proportion of continuous rehabilitation use after six months of the hip fracture surgery. Also, they found that men, higher frailty risk, care dependency before hip fracture surgery, and non-dementia patients were associated with continuity of rehabilitation.

This theme is relevant in the field, but please consider the following comments to improve the readability and to confirm the validity of the study.

Major comments:

1.Page 3, Line 66: Please consider to clarify more details of the study results which supported a minimum requirement of six months rehabilitation after hip fracture surgery in the Japanese guideline. For example, some countries have introduced the reablement interventions which is a multidisciplinary rehabilitation within 3 months after discharge. I am not sure whether all hip fracture patients should take the six months rehabilitation or not, because the intensity of rehabilitation that can be provided in a hospital may not be available or feasible in a nursing home or at home.

2.Page 3, Line 72: Authors mentioned about two previous studies in Japan on the rehabilitation after hip fracture. Please consider the clarification of more details (i.e.,study design, data, limitations etc.) for these studies. Also, authors can mention the other countries’ studies on it. Furthermore, authors are conducting the analysis to identify factors associated with continuity of rehabilitation, so please consider to add what is known in the previous study on it.

3.Page 4, lines 90 – 92: Each data has the different periods for data use. These could make us confused, so please consider to write only actual study period after data management (linkage).

4.Page 4, lines 95 – 97: What is the proportion to what?

5.Page 6, line 126: Please explain the Japanese system of rehabilitation between medical and long-term care insurance. We are not sure why authors must use the both of them.

6.Table S3: Some residents in special nursing home for the elderly (Toku-you in Japanese) have received the rehabilitation. Is it already included in Table S3?

7.Page 6, lines 132-133: Could you clarify the differences between the comorbidities (i.e., dementia and so on) and 20 types of diseases? What is the purpose of describing the 20 types of diseases?

8.Page 6, Lines 139 – 143: Why did authors set the different periods after the surgery to identify the dementia, depression and the frailty risk score, compared to delirium and 20 types of diseases?

9.Page 7, line 153: I am not sure what the convalescent rehabilitation ward is. Please consider the explanation on it.

10.Page 8, line 178: Please consider to add some description for the demographics of the population such as age and sex at least.

11.Table S6 and Table 1 can be combined and shown as a Table 1 because authors are conducting a regression.

12.Figure 2 should be added the value at each month and what is month means at x-axis.

13.Table 2: I’m not sure that the results of univariable analysis is needed because authors did not seem to mention them.

14.Table 2: Who has a responsibility for continuing rehabilitation after hip fracture surgery in Japan? At the acute or sub-acute moments, physicians in hospitals could order it. However, how about in the situation after several months in nursing facilities? Do physicians in clinics do it? Or can care managers in the long-term care insurance system do it?

15.I am worry about the possibility of incorrect and biased estimates because of the nested structure of the population. Especially, if physicians order the rehabilitation, patients are correlated with physicians. Also, physicians are nested by hospitals. Therefore, I think authors should consider this structure in the regression, using the multilevel analysis or cluster SE, even though authors are exploring the related factors simply.

16.Page 13, lines 219 – 229: Authors mentioned the issue around the transition from hospitals to other facilities as a possible reason of the decreasing proportion of rehabilitation over time after the operation. However, it seems to me there are other possible reasons. For example, some patients may be more likely to recover quickly. Other patients may not prefer to continue rehabilitation for whatever reason. Or perhaps there are limited resources for rehabilitation in the home or nursing home. This is a primary outcome in the study, so the authors should consider various possibilities.

17.Page 17, lines 313 - 315: Please clarify the directions of the association. (i.e., positive or negative)

Minor comments:

1.Page 2, Line 43: an intermediate of what?

2.Page 2, Line 45: Please consider to add some odds ratios for care dependency.

3.Table S2 is titled as S3.

6. PLOS authors have the option to publish the peer review history of their article (what does this mean?). If published, this will include your full peer review and any attached files.

Reviewer #1: No

---

## [Author Response · Author response to Decision Letter 0]

6 Jul 2024

July 6, 2024

Dr. Antimo Moretti

Academic Editor

PLOS ONE

Dear Dr. Moretti and Reviewers,

Thank you very much for revising our manuscript entitled “Implementation status of postoperative rehabilitation for older patients with hip fracture in Kyoto City, Japan: A population-based study using medical and long-term care insurance claims data.” We have responded to the editor’s and reviewers’ comments in the point-by-point responses below and revised the manuscript accordingly.

We hope our revised manuscript is now suitable for publication in PLOS ONE. 

Thank you very much for your time and effort in considering our manuscript for publication in PLOS ONE. 

Yours sincerely,

Kosuke Sasaki

Department of Health Informatics, School of Public Health,

Graduate School of Medicine, Kyoto University.

Yoshida-Konoe-cho, Sakyo-ku,

Kyoto 606-8501, Japan.

TEL: +81-75-753-9477

E-mail: sasaki.kosuke.u73@kyoto-u.jp

Reviewers' comments: 

Major comments:

1. Page 3, Line 66: Please consider to clarify more details of the study results which supported a minimum requirement of six months rehabilitation after hip fracture surgery in the Japanese guideline. For example, some countries have introduced the reablement interventions which is a multidisciplinary rehabilitation within 3 months after discharge. I am not sure whether all hip fracture patients should take the six months rehabilitation or not, because the intensity of rehabilitation that can be provided in a hospital may not be available or feasible in a nursing home or at home. 

[Response]

Thank you for the suggestion. We are aware that some countries do not require or continue six months of rehabilitation for various reasons, including differences in healthcare systems and resources. However, Japan has a unique system designed to provide comprehensive rehabilitation through a combination of medical and long-term care insurance. It allows for continuous monitoring even at home and provides both medical and nursing care services.

In Japan, the medical and long-term care insurance systems are designed to ensure that patients receive continuous care. This includes home-based services that support the continuation of rehabilitation beyond acute care hospital settings. Therefore, this study investigated the implementation of rehabilitation services over a 6-month period.

We have rewritten the section to provide more clarity on this matter, as follows:

[Detailed revision]

Clinical practice guidelines from nations including Australia [10], the United Kingdom [11], the United States [12,13], Korea [14], and Japan [15,16] endorse early postoperative rehabilitation for managing hip fractures and continuing post-hospitalization. Although the appropriate duration of rehabilitation varies depending on the healthcare system and the patient's condition, there are reports that rehabilitation for 3 to 6 months is necessary after hip fracture surgeries. A randomized controlled trial of older patients with hip fractures found that outpatient rehabilitation using progressive resistance training for six months after discharge significantly improved walking ability and quality of life compared to low-intensity home exercises [17]. A prospective cohort study found that most aspects of the SF-36, except for the physical role behavior subscale, showed almost complete recovery six months after hip fracture surgeries [18]. Such consistent rehabilitation post-hip surgery enhances physical function [19–21] and quality of life [19,20]. The Japanese guidelines specifically recommend a minimum of six months of postoperative rehabilitation [15,16]. In Japan, the average length of hospital stay is decreasing, making rehabilitation after discharge from acute care hospitals increasingly important [22]. In Japan's multifaceted treatment approach, patient discharge trajectories differ, spanning transfers to other facilities and direct-to-home discharge [23–25]. Uninterrupted rehabilitation is crucial, regardless of transitions in the care setting. (pages 4–5, lines 36–54 in the “Manuscript”)

As you have correctly noted, the appropriate length of rehabilitation may vary depending on the healthcare system and the patient's condition. In this study, it was not possible to determine whether all patients with hip fractures should receive six months of rehabilitation. Noting this as a limitation of the study, we also added the following statement:

“Third, the appropriate duration of rehabilitation may vary depending on the healthcare system and the patient's condition, and this study was not able to determine whether all patients with hip fractures should undergo rehabilitation for six months. However, it does highlight the evidence-practice gap in the implementation of rehabilitation following hip fracture surgeries in Japan.” (page 21, lines 344–349)

2. Page 3, Line 72: Authors mentioned about two previous studies in Japan on the rehabilitation after hip fracture. Please consider the clarification of more details (i.e.,study design, data, limitations etc.) for these studies. Also, authors can mention the other countries’ studies on it. Furthermore, authors are conducting the analysis to identify factors associated with continuity of rehabilitation, so please consider to add what is known in the previous study on it.

[Response]

Thank you for your valuable comments. As per your suggestion, we have provided more details concerning the two previous studies conducted in Japan—including their designs, data, and limitations. We have also mentioned some related studies from other countries and included information regarding the factors associated with rehabilitation continuity from previous research. Below are the original and revised texts for your review:

[Before]

Recent Japanese research indicated that most patients with hip fracture in Japan's acute hospitals promptly undergo rehabilitation.[23] Another study found that intensive rehabilitation in these institutions for dementia-affected patients post-hip surgery enhanced ADL upon discharge.[24] However, there is limited clarity on the sustained implementation of rehabilitation in the medium to long term following discharge. Hence, the present study aimed to explore the rehabilitation implementation status and identify factors influencing sustained rehabilitation for elderly hip fracture patients post-discharge in Japan, a nation confronting super-aging challenges. 

[Detailed revision]

It has been suggested that there are disparities in access to rehabilitation based on patient characteristics. A systematic review describing the equity in randomized controlled trials of rehabilitation interventions following hip fractures revealed that, in > 50% of the 35 trials, potential participants were systematically excluded based on conditions such as residing in nursing homes, cognitive impairment, mobility or functional impairments, age, and ineligibility for surgery [26]. A retrospective cohort study using the census database of all inpatient services provided by public and private hospitals in New South Wales, Australia, reported that older patients, women, those with dementia, those with high frailty risk, and those residing in long-term care facilities were less likely to receive in-hospital rehabilitation [27,28]. Another retrospective cohort study that made use of the Diagnosis Procedure Combination database, which collects inpatient data from over 1500 hospitals across Japan, reported that almost all patients who underwent hip fracture surgeries in acute care hospitals received short-term rehabilitation [29,30]. However, it also reported that those with dementia had fewer opportunities for inpatient rehabilitation after discharge from acute care hospitals [29]. The rate of mid- to long-term extension of rehabilitation after discharge from acute care hospitals for patients following treatments for hip fractures has not been previously reported on, which has raised some concerns surrounding whether sufficient rehabilitation is being provided. Hence, this study aimed to explore the current status of rehabilitation implementation, and identify the factors influencing sustained rehabilitation in older patients with hip fractures post-discharge in Japan—a nation that is currently facing the challenges associated with a rapidly-aging population. (pages 5–6, lines 55–76)

3. Page 4, lines 90 – 92: Each data has the different periods for data use. These could make us confused, so please consider to write only actual study period after data management (linkage).

[Response]

We appreciate your attention to the discrepancy in the duration of data usage. We have revised the manuscript to state that the actual study period was from April 2013 to March 2019 after data management and linkage, to provide greater clarity. The study included patients who underwent surgeries for hip fractures between April 2013 and October 2018 with a follow-up period of 6 months. Therefore, the data-use period was extended to March 2019.

[Before]

In this study, we used the following data of the Medical Care System for the Elderly in the Latter Stage of Life, which is the mandatory medical insurance system for people over 75 years old,[26] basic resident registration (fiscal year [FY] 2010 to 2020), medical insurance claims data (FY 2013 to 2018), LTC insurance claims data (FY 2013 to 2018), and LTC needs certification data (FY 2009 to 2018).

[Detailed revision]

In this study, we used data spanning April 2013 to March 2019 from the following sources: the Japanese “Medical Care System for the Elderly in the Latter Stage of Life” (which is the mandatory medical insurance system for individuals > 75 years of age) [32], basic resident registration, medical insurance claims data, LTC insurance claims data, and LTC needs certification data. These data were merged individually using a unique identifier by Kyoto City and anonymized before we received the data. (page 6, lines 84–90)

4. Page 4, lines 95 – 97: What is the proportion to what?

[Response]

This shows the percentage of data to which personal numbers are assigned in the Kyoto City database according to the data type (e.g., medical and long-term care insurance claims). We have revised the wording to more clearly indicate the percentage of each type of data to which personal identification numbers were assigned.

[Before]

The proportions of personal identification numbers assigned in this database were as follows. The proportion of injury and disease codes of the medical insurance claims data was approximately 99%, the medical care activities codes were about 99%, the LTC insurance claims were about 98%, and the LTC needs certification data was about 87%.

[Detailed revision] 

Personal identification numbers were assigned to ~99% of the medical claims data for injury and illness codes (385,109,527 of 387,547,567 records), ~99% of the medical activity codes (1,237,350,713 of 1,253,415,997 records), ~98% of the LTC insurance claims data (52,387,340 of 53,494,825 records), and ~89% of the LTC care-requiring certification data (861,376 of 964,429 records). (page 6, lines 91–95)

5. Page 6, line 126: Please explain the Japanese system of rehabilitation between medical and long-term care insurance. We are not sure why authors must use the both of them.

[Response]

Thank you for your comments. In Japan, medical and LTC insurance are distinct systems; therefore, claims are filed separately. Therefore, previous studies typically examined either medical or LTC insurance claims in isolation. However, a major strength of our study lies in its integration of medical and LTC insurance claims to provide a comprehensive view of the rehabilitation process. We have added a description of the Japanese medical and long-term care insurance rehabilitation systems to further clarify this. 

[Detailed revision]

The primary outcome was the implementation of rehabilitation for six months after hip fracture surgery as a process measure. We defined the use of rehabilitation as rehabilitation-related claims filed under medical or LTC insurance at least once per month following hip fracture surgery (Tables S2 and S3). In Japan, there are two types of insurance-covered rehabilitation: medical and LTC. Generally, acute and restorative rehabilitation is covered by medical insurance, while chronic rehabilitation is covered by both insurance types. This distinction ensures that patients receive appropriate care tailored to their stage of recovery and long-term needs [24]. To use LTC insurance services, patients must be certified based on their physical and cognitive functions, as well as their care and medical needs. Once certified, patients are required to use LTC insurance rehabilitation services, except during hospitalization or the initial recovery phase [34]. Because this study aimed to investigate the implementation of mid- to long-term rehabilitation after hip fractures, we included both medical and long-term care insurance data. The definition of implementing rehabilitation was defined based on the advice of a physiatrist (MT), physical therapists (KS and HU), and an occupational therapist (TO). (page 8, lines 122–137)

6. Table S3: Some residents in special nursing home for the elderly (Toku-you in Japanese) have received the rehabilitation. Is it already included in Table S3?

[Response]

Thank you for your comments. Unfortunately, our study did not have sufficient data to clarify the implementation status of rehabilitation in specialized nursing homes for older individuals (Toku-you). As was pointed out in Major Comment 15, the implementation of rehabilitation may be influenced by factors related to hospitals and facilities. However, although our database does contain information at both the hospital and facility levels, it has not yet been compiled and maintained in a way that allows for such analyses—thus making it impractical for us to conduct an analysis that considers the nested structure of the population. In response to Major Comment 15, we have added the following text as a limitation of our study.

[Detailed revision]

“Second, this database does not contain information about the physicians who prescribed rehabilitation and information concerning the hospitals or facilities where the patients stayed is not well organized. Therefore, this study could not clarify the relationship between hospitals, facilities, and physicians with the implementation of rehabilitation. Further research is warranted to explore the perspectives of patients and their families on this matter [61,62], as well as facility-level [63] and regional-level factors [64].” (page 21, lines 337–343)

7. Page 6, lines 132-133: Could you clarify the differences between the comorbidities (i.e., dementia and so on) and 20 types of diseases? What is the purpose of describing the 20 types of diseases?

[Response]

Thank you for your valuable comments. Here, we describe the 20 types of diseases specified by the Japanese Ministry of Health, Labor, and Welfare. If a patient has a specified illness, home-visit nursing care for intractable diseases can be used ≥ 4 days per week, 2–3 times per day. If a patient applies for long-term care insurance and is certified as approved for support or long-term care, if the illness specified by the Minister of Health, Labor and Welfare falls under this category the patient will be eligible for home-visit nursing care covered by medical insurance and can use home-visit nursing care 2–3 times per day for ≥ 4 per week. If the patient falls into one of the 20 disease type categories specified by Japan's Ministry of Health, Labor, and Welfare, some rehabilitation claims are made on paper. In such cases, some rehabilitation claims may not have been captured by the database used in this study. To clarify that this study may therefore have undercounted rehabilitation practices, we have revised the wording as follows, in the limitations section.

“Fifth, if a patient had any of the 20 disease types specified by the Japanese Ministry of Health, Welfare and Labor, some rehabilitation claims were made on paper and may therefore not have been captured in the database that was used for this study” (pages 21–22, lines 349–352)

In addition, the following statement was added

---

## [Decision Letter · Decision Letter 1]

3 Sep 2024

Implementation status of postoperative rehabilitation for older patients with hip fracture in Kyoto City, Japan: A population-based study using medical and long-term care insurance claims data

PONE-D-24-11365R1

Dear Dr. Sasaki,

We’re pleased to inform you that your manuscript has been judged scientifically suitable for publication and will be formally accepted for publication once it meets all outstanding technical requirements.

Kind regards,

Masaki Mogi

Academic Editor

PLOS ONE

Additional Editor Comments (optional):

Reviewers' comments:

Reviewer's Responses to Questions

**Comments to the Author**

1. If the authors have adequately addressed your comments raised in a previous round of review and you feel that this manuscript is now acceptable for publication, you may indicate that here to bypass the “Comments to the Author” section, enter your conflict of interest statement in the “Confidential to Editor” section, and submit your "Accept" recommendation.

Reviewer #1: All comments have been addressed

2. Is the manuscript technically sound, and do the data support the conclusions?

Reviewer #1: Yes

3. Has the statistical analysis been performed appropriately and rigorously? 

Reviewer #1: Yes

4. Have the authors made all data underlying the findings in their manuscript fully available?

Reviewer #1: Yes

5. Is the manuscript presented in an intelligible fashion and written in standard English?

Reviewer #1: Yes

6. Review Comments to the Author

Reviewer #1: I appreciate your sincere response to my comments. I have no further comments. I hope this evidence will contribute to better rehabilitation. Great job!

7. PLOS authors have the option to publish the peer review history of their article (what does this mean?). If published, this will include your full peer review and any attached files.

Reviewer #1: No

---

## [Editor Report · Acceptance letter]

4 Sep 2024

PONE-D-24-11365R1 

PLOS ONE

Dear Dr. Sasaki, 

I'm pleased to inform you that your manuscript has been deemed suitable for publication in PLOS ONE. Congratulations! Your manuscript is now being handed over to our production team.

Kind regards, 

on behalf of

Dr. Masaki Mogi 

Academic Editor

PLOS ONE